# Acute Effects of Low Dose of Caffeine Ingestion Combined with Conditioning Activity on Psychological and Physical Performances of Male and Female Taekwondo Athletes

**DOI:** 10.3390/nu14030571

**Published:** 2022-01-28

**Authors:** Ibrahim Ouergui, Nourhene Mahdi, Slaheddine Delleli, Hamdi Messaoudi, Hamdi Chtourou, Zouheir Sahnoun, Anissa Bouassida, Ezdine Bouhlel, Hadi Nobari, Luca Paolo Ardigò, Emerson Franchini

**Affiliations:** 1High Institute of Sport and Physical Education of Kef, University of Jendouba, Kef 7100, Tunisia; ouergui.brahim@yahoo.fr (I.O.); nourhene648@gmail.com (N.M.); bouassida_anissa@yahoo.fr (A.B.); 2High Institute of Sport and Physical Education of Sfax, University of Sfax, Sfax 3000, Tunisia; sdelleli2018@gmail.com (S.D.); hamdimessaoudihamdi@gmail.com (H.M.); h_chtourou@yahoo.fr (H.C.); 3Activité Physique, Sport et Santé, UR18JS01, Observatoire National du Sport, Tunis 1003, Tunisia; 4Pharmacology Department, Faculty of Medicine, University of Sfax, Sfax 3000, Tunisia; zouheir.sahnoun@fmsf.rnu.tn; 5Laboratory of Cardio-Circulatory, Respiratory, Metabolic and Hormonal Adaptations to Muscular Exercise, Faculty of Medicine Ibn El Jazzar, University of Sousse, Sousse 4000, Tunisia; ezdine_sport@yahoo.fr; 6HEME Research Group, Faculty of Sport Sciences, University of Extremadura, 10003 Cáceres, Spain; 7Department of Physical Education and Sports, University of Granada, 18010 Granada, Spain; 8Department of Exercise Physiology, Faculty of Educational Sciences and Psychology, University of Mohaghegh Ardabili, Ardabil 56199-11367, Iran; 9Sports Scientist, Sepahan Football Club, Isfahan 81887-78473, Iran; 10Department of Neurosciences, Biomedicine and Movement Sciences, School of Exercise and Sport Science, University of Verona, Via Felice Casorati 43, 37131 Verona, Italy; 11Martial Arts and Combat Sports Research Group, School of Physical Education and Sport, University of São Paulo, São Paulo 05508-030, Brazil; efranchini@usp.br

**Keywords:** combat sports agility, ergogenic aid, plyometric

## Abstract

This study investigated low-dose caffeine ingestion, conditioning activity (CA) effects on psycho-physical performances in young taekwondo athletes. In a randomized, double-blind, counterbalanced, crossover design, 20 athletes (10 males; 17.5 ± 0.7 yrs) performed taekwondo-specific agility test (TSAT), 10 s/multiple frequency speed of kick test (FSKT-10s/FSKT-mult) after ingesting 3 mg·kg^−1^ caffeine (CAF) or placebo (PL) 60 min before performing standard warm-up without (NoCA) or with CA (3 × 10 vertical jumps above 40 cm), resulting in four experimental (PL + NoCA, CAF + NoCA, PL + CA, and CAF + CA) and one control (warm-up session without CAF or CA) conditions. Mood/physical symptoms (MPSS), subjective vitality (SVS), and feeling (FS) scales were analyzed post-to-pre for all conditions. Ratings of perceived-exertion and perceived-recovery status were determined after tests. For TSAT, CAF + CA induced better performance compared with all conditions (*p* < 0.001). For FSKT-10s and FSKT-mult, CAF + CA induced better performance compared with all conditions (*p* < 0.001). For MPSS, FS, CAF + NoCA induced higher scores than PL + NoCA and PL + CA (*p* = 0.002, 0.009 for MPSS; *p* = 0.014, 0.03 for FS). For SVS, PL + CA elicited lower scores than PL + NoCA and CAF + NoCA (*p* = 0.01, 0.004). Sex comparisons resulted in better performances for males for TSAT (*p* = 0.008), FSKT-10s (*p* < 0.001), FSKT-mult (*p* < 0.01), MPSS (*p* = 0.02), SVS (*p* = 0.028), and FS (*p* = 0.020) scores. Caffeine and conditioning activity are two efficient performance-enhancing strategies, which could synergistically result in greater psycho-physical performances.

## 1. Introduction

Taekwondo is a combat sport characterized by brief and intense actions interspersed by long, low-intensity movements [1]. The high-intensity actions performed during 2 min rounds require a high level of agility, quick reactions, and a high capacity to kick powerfully at a limited scoring area and under strict time constraints [2]. To prepare taekwondo athletes to handle the competition demands, coaches and sport scientists are increasingly employing different performance optimization strategies. Among the most proposed strategies in the field of athletic performance is the use of caffeine (CAF) as an ergogenic aid, which has been widely introduced to improve explosive exercise performances [3]. The ergogenic potential of CAF was attributed to both central and peripheral mechanisms [4,5]. In fact, CAF exerts its effect via blocking the adenosine A_1_ and A_2a_ receptors, which reduces fatigue level, improves neurotransmission, and increases motor unit firing rates [5]. In the periphery, CAF enhances muscle contraction throughout calcium ion mobilization [6] and phosphodiesterase inhibition [4].

In combat sports, there is no consensus about CAF effects on physical performances [7]. In fact, some investigations reported improvements [8,9,10,11], while others did not [12,13]. This difference could be related to the methodological and the inter-individual variations. In fact, the ergogenic effects of CAF are influenced by participants’ characteristics (i.e., sex, age, training status), CAF supplementation protocol (i.e., dose, form, ingestion time, time of day of CAF supply), tolerance to CAF, genetics, and nature of exercise [3,7,14,15,16,17].

While CAF alone offers an attractive and useful tool to improve physical performance, as well as physiological and psychological responses, there are other strategies used by coaches to enhance muscle performance and may be synergistic when combined with CAF [18]. In fact, the post-activation performance enhancement (PAPE), previously denominated as post-activation potentiation (PAP) in classical investigations [19,20], is the stimulus by which muscular performance is enhanced in response to a voluntary conditioning activity (CA) [19,20]. Indeed, PAPE may occur when the potentiation effects are higher than the neuromuscular fatigue generated by the CA [19]. Since PAP and PAPE were distinguished as two different phenomena [20], PAPE has been explained by increased muscle temperature, intramuscular fluid accumulation, and increased neural drive [19]. To generate a PAPE stimulus, different modes of CA have been used [21]. However, plyometric exercises were suggested to be better than traditional resistance CAs [22]. This was reasonable as plyometric exercises may cause less fatigue than traditional resistance exercise, allowing stronger enhancement effects and shorter time to achieve it [22]. In combat sports, there were contradictory results, with some studies revealing performance enhancement following plyometric CA [23,24,25], whereas others did not [26,27].

To our current knowledge, only one study [18] investigated the effects of PAP, with and without prior 5 mg/kg CAF ingestion on countermovement jump performance in male soccer players and showed that performance was better in 1, 3, and 5 min after the conditioning activity in CAF condition, whereas in the placebo condition, performance improved only in the 5-minute time-point. However, studies in this field of investigation combat sports, especially within taekwondo practitioners, are limited. Moreover, the large inter-individual response variation was considered as a common reason for the contradictory results among studies [18]. There are few investigations that compared performances between sexes in both CAF [28,29,30] and CA [31] interventions and showed that males were more responsive than females due to hormonal variation [32], menstrual cycle phases [33], the consumption of oral contraceptive which extended caffeine half-life and prolonged responses [10], difference in muscle typology (i.e., higher type II fiber in males than in females) [31], and body composition [29].

From a psychological side, CAF has been reported to enhance positive mood components (i.e., vigor and tension) and increase the subjective vitality profile [34,35], reduce fatigue, and increase the feeling of pleasure even when energy levels are low [36]. In taekwondo, like other sports, the ability to reduce fatigue and present more positive psychological responses could result in greater performances [1]. However, the effectiveness of low doses of CAF on these psychological aspects was not examined in taekwondo athletes, especially when it is combined with a CA. Therefore, the present study investigated the effects of using a plyometric CA combined with a low dose of caffeine ingestion on physical performances and psychological responses of young male and female taekwondo athletes. Because caffeine supplementation has been reported to reduce fatigue [4] and CA has been proposed as an efficient strategy to enhance performance [19,20], it was hypothesized that the combination might cause better improvements compared with the isolated conditions. Moreover, as the impact of CA is greater in stronger subjects [22], and CAF metabolism is greater in males than in females [17], it was hypothesized that males would present better performances than females.

## 2. Materials and Methods

### 2.1. Participants

A priori power analysis was calculated using the G*Power software (Version 3.1.9.4, University of Kiel, Kiel, Germany) using the F test family (repeated measures, within-between interaction), with five conditions. The analysis revealed that a total sample size of 15 would be sufficient to find significant and large-sized effects of condition (effect size f = 0.7, α = 0.05) with an actual power of 95%. Because some athletes could drop out of the study, twenty taekwondo athletes (10 males and 10 females; age: 17.5 ± 0.7 years; body mass: 59.2 ± 10.0 kg; height: 168 ± 9 cm) with at least 6 years of experience volunteered to participate in this study. All participants were non-smokers, and according to the classification proposed by Filip et al. [37], they were moderate CAF consumers (less than 3 cups of coffee per day), as one cup of coffee is assumed to contain 100 mg of CAF [38]. None of the athletes was suffering from any restrictions to sport practice (e.g., cardiovascular diseases), they did not use dietary supplements or anti-inflammatory drugs, and they had not taken psychotropic drugs in the 3 months prior to the study. For females, they were selected as non-contraceptive pills users, as these medications may interfere with caffeine pharmacokinetics [10]. Moreover, the menstrual cycle was considered in the organization of the experimental sessions (i.e., all female athletes were instructed to participate in the testing during their early follicular phase). The participants were asked to follow the same diet, avoid alcoholic substances and vigorous exercise, and restrain from CAF consumption (in drinks and supplements) during the 48 h prior to each session. Before taking part in the experimentation, all participants were informed about the procedures, the possible risks, and the discomforts involved in the investigation, and they and/or their parents signed an informed consent form. This study was conducted in accordance with the Declaration of Helsinki, and the protocol was fully approved by a local research ethics committee (CPP SUD N° 0332/2021).

### 2.2. Experimental Design

This study followed a double-blind, counterbalanced, crossover design to investigate the effects of CAF supplementation and CA during the warm-up on the subsequent 10 s frequency speed of kick test (FSKT-10s), multiple frequency speed of kick test (FSKT-mult), and taekwondo-specific agility test (TSAT) performances, the Mood and Physical Symptoms Scale (MPSS), the Subjective Vitality Scale (SVS), the Feeling Scale (FS), the rating of perceived exertion (RPE) and perceived recovery status (PRS).

During the first visit, familiarization about the procedures and tests was conducted, and athletes’ anthropometrics characteristics were determined (72 h before the experiments). For the test sessions, athletes were submitted to five conditions (4 experimental conditions and 1 control). In fact, in a double-blind fashion, athletes ingested a 3 mg/kg of CAF or placebo (PL) diluted in 200 mL of water. The 3 mg/kg of caffeine was chosen as it was considered a safe dose, and its effectiveness was previously reported in combat sports [39]. One hour after supplementation, the athletes executed 2 warm-ups protocols, consisting of a standard warm-up (i.e., without CA (NoCA), 10 min running at 9 km/h followed by 2 min rest), or warm-up session followed by a plyometric conditioning activity protocol (i.e., 3 sets of 10 vertical jumps above 40 cm). During the control session, athletes were submitted only to a standard warm-up session, in which athletes performed only 10 min of running at 9 km/h followed by 2 min of rest. During conditions in which a CA was performed, 10 min of rest after the physical tests was attributed. This post-stimulus recovery period was chosen based on the results of a previous meta-analysis [40], which showed that 7–10 min was an optimal recovery period to maximize the effects of potentiation. Furthermore, the Physical Mood and Symptoms Scale (MPSS), the Subjective Vitality Scale (SVS), and the Feeling Scale (FS) were completed before and 1 h post-supplementation. The 1 h duration was used since it was considered as an optimal duration for full absorption of caffeine and allowing the peak concentration of caffeine to be observed [41]. Therefore, athletes performed the specific tests through five different conditions: (1) control condition (no supplementation + NoCA), (2) placebo without CA (PL + NoCA), (3) caffeine without CA (CAF + NoCA), (4) placebo with CA (PL + CA), and (5) caffeine with CA (CAF + CA). To compare the effects of different conditions on perceived exertion and recovery, athletes were asked to rate their perceived exertion and recovery after finishing the testing procedure (Figure 1). The sessions were separated by an interval of seven days to allow sufficient recovery between sessions and to ensure caffeine elimination [41]. For each experimental session and to avoid identification, supplements were administered in opaque, unmarked containers and handled by a trained person. Participants were instructed to not discuss or compare tastes or make assumptions about what they had ingested and were supervised by staff to ensure that they drank the full amount, and no exchanges were allowed. In addition, to check the success of blinding, each subject was asked to identify what he/she took, and any of the athletes distinguished correctly which supplement had been ingested. All the sessions were conducted at the same time of day (from 10 a.m. to 12 p.m.) to overcome the time-of-day effects.

### 2.3. Testing Procedure

#### 2.3.1. Taekwondo-Specific Agility Test

The athlete began the test from a guard position with both feet behind the start/finish line. At his/her discretion, the athlete moved as quickly as possible towards the center point. Then, following his/her own preference, he/she turned towards partner 1 by performing a sideways movement and performed a roundhouse kick lead leg. Subsequently, he/she turned and shifted to partner 2 and performed a right another roundhouse kick with the other lead leg. Next, he/she returned to the center, moved forward to partner 3 in a guard position, and performed a double roundhouse kick. Finally, the athlete retreated to the start/finish line [42]. The completion time was measured using photocells (Brower Timing Systems, Salt Lake City, UT, USA). Three trials were performed by each athlete, and the best one was used for analysis. The intra-class correlation coefficient (ICC) for test–retest in the present study was 0.98.

#### 2.3.2. 10 s Frequency Speed of Kick Test

The test was performed as described by da Silva Santos et al. [27]. The athlete had to perform the maximum number of kicks against a punching bag by alternating the right and left leg. The technique used during the test was the Bandal Chagui. The number of techniques performed during the 10 s of the test represented the performance index [27]. The ICC for the test–retest in the present study was 0.95.

#### 2.3.3. Multiple Frequency Speed of Kick Test

The same procedures in the FSKT-10s were adopted for the FSKT-mult. The athlete performed five sets of FSKT-10s with a 10 s rest interval between repetitions. Performance was determined by the total number of kicks performed in each set and the total number of kicks in 5 sets, which allowed determining the kick decrement index (DI), which is calculated as follows [43] (Equation (1)):DI (%) = [1 − ((FSKT1 + FSKT2 + FSKT3 + FSKT4 + FSKT5)/(Best FSKT set × Numbers of sets))] × 100(1)

The ICC for the test–retest in the present study was 0.85.

#### 2.3.4. Mood and Physical Symptoms Scale

The questionnaire consists of 20 items answered on a scale from 1 to 5 with 1 = strongly disagree and 5 = strongly agree, describing feelings of mood, including alertness and various physical sensations. These 20 points were reduced to a smaller number of variables for analysis as follows: “alertness” (I feel energetic, I feel very tired, I can think clearly, I don’t feel like doing much, I feel very awake, I have trouble thinking clearly, I feel like doing something active, I feel sleepy), “happy/friendly” (I feel happy, I feel friendly, sad, angry, happy), “headache” (I have a headache), and “other negative physical symptoms” (my muscles hurt, I feel strong, my stomach hurts, my fingers are tingling, I feel healthy, I feel good) [44]. The MPSS showed good reliability, with Cronbach’s alpha of 0.78 [44].

#### 2.3.5. Subjective Vitality Scale

A simple self-report scale designed to subjectively measure the level of vitality was used. Participants were asked to indicate their agreement with 7 statements related to subjective feelings of energy and vitality using a 7-point Likert scale, where 1 = not at all true and 7 = very true. The 7 items were: (1) I feel alive and vital, (2) I don’t feel very energetic, (3) sometimes I feel so alive I just want to burst, (4) I have energy and spirit, (5) I look forward to each new day, (6) I nearly always feel alert and awake and (7) I feel energized. The possible range of a score was from 7 to 49, with higher scores indicating stronger subjective vitality [45]. The SVS showed good reliability, with Cronbach’s alphas of 0.84 and 0.86 for two samples [45].

#### 2.3.6. Feeling Scale

The FS was used to assess general affective valence (pleasure and dissatisfaction), described by Hardy and Rejeski [46]. Participants indicated their current feelings on an 11-point bipolar scale ranging from +5 to −5. The language anchors were very good (+5), good (+3), fairly good (+1), neutral (0), fairly bad (−1), bad (−3), and very bad (−5). The FS has been shown to be related to other measures of affective valence and present and past physical activity participation [46].

#### 2.3.7. Rating of Perceived Exertion

Perceived exertion was assessed using the 0–10 exertion perception scale adapted by Foster et al. [47]. This is a scale ranging from 0 to 10, with corresponding verbal expressions, that gradually increases with the intensity of perceived sensation (0 = Nothing at all; 1 = Very light; 2 = Light; 2–3 = Moderate; 4–5 = Somewhat heavy; 6–7 = Heavy; 8–9 = Very heavy, and 10 = Very, very heavy).

#### 2.3.8. Perceived Recovery Status

The PRS was used to assess the athlete’s level of recovery. All participants complete a self-assessment report in the form of an 11-point representation scale from 0 (very little recovery/extremely tired) to 10 (very well recovered/very energetic) [48].

### 2.4. Statistical Analysis

The statistical analysis was performed using SPSS 20.0 statistical software (IBM corps., Armonk, NY, USA). Data are presented as mean and standard deviation. The Kolmogorov–Smirnov test was used to check and confirm the normality of data sets, and the Levene test was used to verify the homogeneity of variances. Sphericity was tested using the Mauchly test. A two-way analysis of variance (ANOVA) (condition × sex) with repeated measurements was used to compare TSAT, FSKT-10s, RPE, and RPS throughout the different experimental conditions, while the FSKT-mult outcomes (total number of techniques and decrement index) were compared using a multivariate analysis of variance. Moreover, a three-way ANOVA (condition × time × sex) was performed to compare MPSS, SVS, and FS, and partial eta squared (η_p_^2^) effect size values were reported and classified as 0.01 = small, 0.09 = medium, and 0.25 = large [49]. When the ANOVA indicated a significant difference, Bonferroni was used as a post hoc test. Due to the substantial number of results, only significant differences were reported. Standardized effect size analysis (Cohen’s d) was used to interpret the magnitude of differences between variables and considered as trivial (≤0.20), small (≤0.60), moderate (≤1.20), large (≤2.0), very large (≤4.0), and extremely large (>4.0) [50]. In addition, the upper and lower 95% confidence intervals of the difference (95% CI) were calculated for the corresponding variation. The level of statistical significance was set at *p* ≤ 0.05.

## 3. Results

Table 1 presents the physical performances recorded during different conditions.

### 3.1. Taekwondo-Specific Agility Test

The results showed a significant condition effect (F_4,36_ = 35.376; *p* < 0.001; η_p_^2^ = 0.96), with CAF + CA condition resulting in better performance compared with PL + NoCA (95% CI = from −0.913 to −0.403; d = −1.46; *p* < 0.001), CAF + NoCA (95% CI = from −0.579 to −0.198; d = −0.96; *p* < 0.001), PL + CA (95% CI = from −0.435 to −0.073; d = −0.63; *p* = 0.006), and the control (95% CI = from −0.931 to −0.443; d = −1.95; *p* < 0.001) conditions. Moreover, PL + CA condition elicited better performances compared with PL + NoCA (95% CI = from −0.596 to −0.239; d = −0.81; *p* < 0.001), CAF + NoCA (95% CI = from −0.227 to −0.042; d = −0.30; *p* = 0.004), and control (95% CI = from −0.646 to −0.220; d = −1.06; *p* < 0.001) conditions. In addition, CAF + NoCA resulted in lower TSAT time in comparison with PL + NoCA (95% CI = from −0.449 to −0.090; d = −0.54; *p* = 0.004) and the control (95% CI = from −0.501 to −0.096; d = −0.73; *p* = 0.004) conditions. Similarly, there was a significant sex effect (F_1,9_ = 11.398; *p* = 0.008; η_p_^2^ = 0.559), with lower TSAT time in males compared with females (95% CI = from −1.220 to −0.241; d = −1.70; *p* = 0.008).

### 3.2. Frequency Speed of Kick Test

There was a significant condition effect (F_4,36_ = 79.844; *p* < 0.001; η_p_^2^ = 0.982), with CAF + CA condition resulting in better performance compared with PL + NoCA (95% CI = from 3 to 5; d = 2.5; *p* < 0.001), CAF + NoCA (95% CI = from 2 to 4; d = 1.97; *p* < 0.001), PL + CA (95% CI = from 1 to 3; d = 1.53; *p* < 0.001), and control (95% CI = from 3 to 6; d = 3.58; *p* < 0.001) conditions. Moreover, PL + CA condition resulted in better performance compared with PL + NoCA (95% CI = from 1 to 2; d = 1.12; *p* < 0.001), and the control (95% CI = from 1 to 4; d = 1.89; *p* = 0.001) conditions. In addition, CAF + NoCA condition resulted in better performance compared with PL + NoCA (95% CI = from 0.3 to 2; d = 0.67; *p* = 0.005) and control (95% CI = from 0.5 to 3; d =1.34; *p* = 0.007) conditions. Similarly, results showed a significant sex effect (F_1,9_ = 28.551; *p* < 0.001; η_p_^2^ = 0.76), with better performance in males than females (95% CI = from 2 to 4; d = 1.78; *p* < 0.001).

### 3.3. Multiple Frequency Speed of Kick Test

#### 3.3.1. Total Number of Techniques

There was a significant condition effect (F_4,90_ = 28.984; *p* < 0.001; η_p_^2^ = 0.563), with a higher total number of techniques in the CAF + CA condition compared with PL + NoCA (95% CI = from 11 to 24; d = 1.78; *p* < 0.001), CAF + NoCA (95% CI = from 6 to 19; d = 1.61; *p* < 0.01), PL + CA (95% CI = from 4 to 16; d = 1.52; *p* < 0.001), and the control (95% CI = from 16 to 29; d = 3.7; *p* < 0.001) conditions. Moreover, compared with control condition, the total number of techniques was significantly higher in the PL + CA (95% CI = from 6 to 19; d = 2.36; *p* < 0.001) and CAF + NoCA (95% CI = from 3 to 16; d = 1.67; *p* < 0.001) conditions. Likewise, PL + CA condition induced higher values compared with PL + NoCA condition (95% CI = from 1 to 14; d = 1.20; *p* = 0.010). Furthermore, a significant sex effect was found (F_1,90_ = 66.815; *p* < 0.001; η_p_^2^ = 0.426), with better performance in males than females (95% CI = from 9 to 14; d = 1.83; *p* < 0.001). Moreover, there was an interaction effect between condition and sex (F_4,90_ = 3.627; *p* = 0.009; η_p_^2^ = 0.414), with females in CAF + CA eliciting higher values compared with PL + NoCA and control conditions (95% CI = both from 5 to 24; both d = 2.49; both *p* < 0.001) and males in CAF + CA resulting in higher values compared with PL + NoCA (95% CI = from 5 to 24; d = 3.28; *p* < 0.001), CAF + NoCA (95% CI = from 5 to 24; d = 2.34; *p* < 0.001), PL + CA (95% CI = from 5 to 24; d = 2.31; *p* < 0.001), and control (95% CI = from 5 to 24; d = 4.96; *p* < 0.001) conditions, and control condition for males eliciting lower values compared with PL + NoCA (95% CI = from −19 to −1; d = −1.61; *p* = 0.023), CAF + NoCA (95% CI = from −23 to −5; d = −2.01; *p* < 0.001), and PL + CA (95% CI = from −28 to −10; d = −3.73; *p* < 0.001) (Table 1). Moreover, males elicited higher values compared with females in PL + NoCA (95% CI = from 6 to 18; d = 2.06; *p* < 0.001), CAF + NoCA (95% CI = from 4 to 17; d = 1.23; *p* < 0.001), PL + CA (95% CI = from 8 to 21; d = 2.16; *p* = 0.001), and CAF + CA (95% CI = from 12 to 25; d = 2.96; *p* < 0.001) conditions.

#### 3.3.2. Decrement Index

There was a significant condition effect (F_4,90_ = 17.859; *p* < 0.001; η_p_^2^ = 0.442), with lower DI in CAF + CA condition compared with PL + NoCA (95% CI = from −7 to −2; d = −2.46; *p* < 0.001), CAF + NoCA (95% CI = from −5 to −0.01; d = −1.07; *p* = 0.049), and the control (95% CI = from −9 to −4; d = −2.56; *p* < 0.001) conditions. Moreover, compared with control condition, DI was significantly lower in the PL + NoCA (95% CI = from −7.987 to −0.881; d = −0.95; *p* = 0.013), CAF + NoCA (95% CI = from −6.859 to −1.833; d = −1.54; *p* = 0.001), and PLA + CA (95% CI = from −7.987 to −0.881; d = −1.42; *p* = 0.013) conditions. Furthermore, a significant sex effect was found (F_1,90_ = 4.990; *p* < 0.001; η_p_^2^ = 0.053), with better performance in males than females (95% CI = from 10 to 17; d = 1.83; *p* < 0.001).

Table 2 presents the psychological responses, and Table 3 shows perceived exertion and recovery scores through different conditions.

### 3.4. Mood and Physical Symptoms Scale

Results showed that there was a significant condition effect (F_1_._365,12_._287_ = 14.427; *p* = 0.002; η_p_^2^ = 0.861), with CAF + NoCA resulting in higher scores compared with the PL + NoCA (95% CI = from 2 to 8; d = 0.34; *p* = 0.002) and the PL + CA (95% CI = from 2 to 14; d = 0.61; *p* = 0.009). Similarly, there was a significant time of measurement effect (F_1,9_ = 10.829; *p* = 0.009; η_p_^2^ = 0.546), with higher values in post- than pre-conditions (95% CI = from 1 to 5; d = 0.54; *p* = 0.009). There was an interaction effect between condition, sex, and time of measurement (F_1_._461,13_._149_ = 5.485; *p* = 0.004; η_p_^2^ = 0.379), with males eliciting higher values post compared with pre for PL + NoCA and CAF + NoCA conditions (95% CI = from 1 to 14 and 1 to 12; d = 0.28 and 1.208; *p* = 0.024 and 0.034, respectively). Moreover, females presented higher values post compared with pre for PL + CA and CAF + CA conditions (95% CI = from 1 to 12 and from 1 to 8; d = 0.65 and 0.72; *p* = 0.038 and 0.036, respectively). Additionally, females elicited higher values post CAF + NoCA compared with PL + NoCA and PL + CA (95% CI = from 1 to 12 and from 5 to 16; d = 0.716 and 0.43; *p* = 0.01 and 0.001, respectively) (Table 2).

### 3.5. Subjective Vitality Scale

There was a significant condition effect (F_3,27_ = 10.313; *p* = 0.006; η_p_^2^ = 0.815), with PL + CA condition resulting in lower scores than PL + NoCA (95% CI = from −6 to −1; d = −0.2; *p* = 0.01), and CAF + NoCA (95% CI = from −9 to −2; d = −0.67; *p* = 0.004) conditions. In addition, there was a significant sex effect (F_1,9_ = 6.889; *p* = 0.028; η_p_^2^ = 0.434), with higher values for males than females (95% CI = from 0.4 to 5; d = 0.72; *p* = 0.028). There was a significant interaction effect between sex and condition (F_3,27_ = 4.743; *p* = 0.041; η_p_^2^ = 0.67), with higher scores for males than females in CAF + NoCA and CAF + CA conditions (95% CI = from 2 to 9 and from 0.004 to 7; d = 1.015 and 1.52; *p* = 0.007 and 0.050, respectively).

### 3.6. Feeling Scale

Results showed a sex effect (F_1,9_ = 7.494; *p* = 0.020; η_p_^2^ = 0.469), with males eliciting higher scores than females (95% CI = from 0.1 to 1; d = 0.33; *p* = 0.020). Moreover, there was a significant interaction effect between condition and sex (F_3,7_ = 8.207; *p* < 0.001; η_p_^2^ = 0.477), with males eliciting higher values than females in CAF + NoCA and CAF + CA (95% CI = from 0.1 to 2 and from 1 to 3; d = 0.14 and 0.93; *p* = 0.037 and *p* < 0.001, respectively).

## 4. Discussion

The purpose of this study was to investigate the effects of combining CAF and CA on psychological responses and specific physical performances in young taekwondo athletes. The present study showed that combining CAF with plyometric exercise during warm-up improved physical performance in all specific tests (i.e., TSAT, FSKT-10s, and FSKT-mult (total number of techniques)) compared with the other conditions. Moreover, males elicited greater physical and psychological responses compared with females. Furthermore, the CAF + NoCA condition resulted in more positive psychological responses (i.e., higher MPSS and SVS scores) than the placebo conditions. However, RPE and PRS did not differ across conditions.

The present study showed better physical performances recorded during TSAT, FSKT-10s, and FSKT-mult following CAF supplementation and CA. It seems difficult to compare our results to other studies since no previous research has examined the acute effects of CAF and CA combination in combat sports. However, to our current knowledge, there was only one previous study [18] that has examined the combined effects of CAF and plyometric exercises in the male soccer players’ performance. This study showed that the combined condition resulted in greater jumping performance following plyometric and sled towing stimuli with 5 mg/kg of CAF. Despite the difference between our study and the aforementioned one in terms of participants, testing procedure, and sport modality, its findings support our results because lower body performance’s improvement needs greater explosive power [3].

Regarding the effectiveness of CAF and CA separately used, our results are in agreement with those reported in previous studies in combat sports. In fact, the effectiveness of plyometric-based CA to improve subsequent physical performance has been confirmed in previous studies investigating other combat sports [23,24,25]. In karate athletes, Margaritopoulos et al. [24] showed that three sets of five tuck jumps followed by 5 min interval induced a 3.5% improvement in jump height. In judo athletes, similar results were found by Miarka et al. [25], who reported a 12% increase in the number of throws during the special judo fitness test when athletes were submitted to ten sets of three consecutive jumps followed by 3 min of rest. However, da Silva Santos et al. [27] did not report improvement in kicking performance during the FSKT-10s following three sets of ten vertical jumps above 40 cm barrier even when 10 min of rest was given. In addition, Castro-Garrido et al. [26] did not find significant differences in the total number of kicks during the FSKT-mult 10 min after performing three sets of ten jumps above 20 cm. This contradiction could be related to the difference in the methodological approaches from one side and to the inter-individual variation from the other side. Moreover, in taekwondo, speed is a key element in the Bandal Tchagui technique [27]. Since the rectus femoris muscle is the most recruited muscle in this technique [2], the performance enhancement following the plyometric CA was mainly located in the quadriceps muscles [51]. The main mechanism under this enhancement is probably due to a reduction in the pennation angle, which improves force transmission to the tendon [18].

Furthermore, the physical improvements following CAF supplementation without CA were similar to those reported in previous studies. In fact, with male taekwondo athletes, it was reported that 5 mg/kg of CAF ingestion was effective in enhancing reaction time 1 h following supplementation [9,11]. Using a similar dose of CAF (i.e., 3 mg/kg), Diaz-Lara et al. [39] reported that maximal lower body strength production was improved in elite Brazilian jiu-jitsu athletes. In the present study, the impact of CAF on physical performances may be explained by both peripheral and central mechanisms. More specifically, at the peripheral level, CAF increased the bioavailability of calcium in the myoplasm [6], which positively affected the energy release by the muscle, explaining then the ergogenic effects observed in both agility and kicking performances. Furthermore, CAF serves as an adenosine antagonist at the central level, boosting neurotransmitter production and nervous system activation [5].

Likewise, scoring more points is the main goal in taekwondo competitions [43]. To achieve this aim, elite taekwondo competitors need to perform strong and fast kicks [2] with less exhaustion [43]. However, in repeated high-intensity actions, fatigue has a detrimental influence on kick time and impact [52]. In the present study, the decrease in fatigue rate indicated by the DI could confirm the effectiveness of the plyometric CA on taekwondo-specific performance. This result was not in accordance with those previously reported in taekwondo athletes [26,43]. The discrepancy could be related to the difference in the methodological approaches (i.e., the CA mode and the jump height used in each trial). For the CAF effect, it has been argued that the most pronounced effects of CAF are evident in delaying the onset of fatigue [9]. This was confirmed in our study as CAF supplementation decreased the decrement index. Similar results were reported by the study of Astley et al. [8], which assessed the influence of acute ingestion of 4 mg/kg of CAF on SJFT and showed an increase in performance 60 min post-supplementation by achieving a greater number of throws (31%) and reducing the fatigue index by 22.29%. Moreover, in taekwondo, 5 mg/kg of CAF was sufficient to improve performance in a specific task and effective in delaying fatigue during successive taekwondo bouts [11] and after strenuous tasks [9]. Indeed, Lopes-Silva et al. [13] reported increased glycolytic activation during taekwondo simulated match after the ingestion of 5 mg/kg of CAF 1 h before the task. Thus, a higher glycolytic activation may also be a contributing factor to improved performance in our taekwondo-specific tests.

CAF and CA are two different conditioning strategies, and there is a common explanation of performance enhancement following their combined application. In fact, the decrease in the fatigue level could be attributed to an enhancement in neuromuscular efficiency following CAF supplementation [9] and CA application [53], which increased the recruitment of high-threshold motor units [21] or even increased the activation of synergists [54], resulting in greater intra- and inter-muscle coordination [55,56].

It is well known that the ergogenic potential of CAF is established by its direct action on the central nervous system, leading to improved alertness and reaction time and reducing the rating of perceived exertion [57]. Although physical performances were increased after the combined use of CA and CAF, no differences between conditions were found for RPE and PRS scores. This could indicate that all experimental conditions resulted in similar perceived exertion and recovery. This result was similar to those reported in previous studies in combat sports, which revealed the lack of significant caffeine effect [11,13] and CA [27,43] on these parameters. The fact that all our tasks involved all-out efforts is also a fact to explain the absence of difference between conditions regarding RPE. However, other interventions with CAF ingestion [8,58] or after performing a CA [23] showed a reduced RPE in both male and female athletes.

Considering the psychological responses, MPSS, FS, and SVS scores were higher after caffeine ingestion compared with the placebo condition. These results are in accordance with those previously reported [34,35], showing that 6 mg/kg of CAF ingestion 60 min prior to testing enhanced positive mood components (i.e., vigor and tension) and increased subjective vitality profile in elite athletes. These results suggest that CAF intake leads to the finest emotional state to perform a physical task and prepares the athlete to face a severe mental effort [34]. The effects of CAF on these parameters may be related to the enhancement of brain activation [34]. Since that adenosine formation is primarily regulated by energy deprivation, the caffeine antagonism of adenosine signaling can override this regulation, resulting in fatigue reduction and increased pleasure even when energy levels are low [36].

The current study showed that males elicited significantly better performances than females. This result is in accordance with previous studies [17,28], which reported that the ergogenic aid of caffeine declines within females in anaerobic exercise tests. In fact, it was shown that the improvement in strength and power performance is more pronounced in males compared with females [17]. However, it was not the case for isometric contractions, where the ergogenic effect of CAF (i.e., 6 mg/kg) on muscle power and muscle endurance did not show a sex difference [6]. In this context, it was expected that CYP1A2 and ADORA2A enzymes modulate the rate of CAF metabolism, which determines the genotype characteristics and causes different responses to CAF [16]. However, even though they have the same genotypes, it has been suggested that females require a longer time to metabolize CAF than males [15]. This fact could be explained by the greater half-life of CAF in females than in males [14]. Moreover, the effect of CAF was reported to be greater in individuals with larger muscle mass [59]. Therefore, the higher performances recorded by males than those by females could be explained by this fact, as males are stronger than females [58]. Supporting our findings, previous studies showed that, with the same dose of CAF, males had greater ergogenic impact than females, particularly regarding power and speed [29]. Concerning the impact of sex on psychological responses, results showed that more positive affect and vitality profiles were presented by males than females in CAF + NoCA and CAF + CA conditions. In this consideration, studies applying neuro- and psycho-pharmacological approaches suggested that males perceive lower somnolence and greater activation after supplementation compared with females [30]. Therefore, in line with what was recommended by Martins et al. [14], it is important to be cautious when extending male findings to females.

The improvement of physical performances and psychological responses following the combined conditions was not clear. However, the synergistic effect of CA and CAF ingestion suggested by Guerra Jr et al. [18] could explain our results. In fact, the positive action of CAF on MPSS and VS could improve the athletes’ behaviors to exercise, which probably improves alertness and reaction time [57]. Furthermore, it was revealed that the effects of caffeine on cognition and brain activation are greater with low doses of caffeine than with moderate and high doses [60]. Thus, the 3 mg/kg ingested in the present study could be efficient to block the inhibitor effects of adenosine. At this time, the CA prepared the involved muscles for high-intensity movements through a variety of processes, which can include those mentioned above. Consequently, the reported outcomes might be the result of a mix of central and peripheral mechanisms. However, this explanation remains a theory that must be tested in future research.

Although this study led to a novel approach to enhance taekwondo athletes’ performance, some limitations should be acknowledged. In fact, daily CAF consumption was not measured using objective markers (i.e., urine caffeine production, plasma caffeine, or caffeine metabolite levels). In addition, there has been no research into the mechanisms underlying the synergistic effects of CAF and CA, and genetic polymorphisms were not controlled to explain the results. Finally, the tests used are specific and could mimic the competition demands, but they did not provide information about athletes’ tactical behaviors. Therefore, it is of great importance to investigate the effects of CAF and CA combination on taekwondo simulated competition.

## 5. Conclusions

Caffeine supplementation and conditioning activity are two effective strategies aiming to improve the physical performances and the psychological state of athletes. The present study showed the effectiveness of combining the effects of the low dose of caffeine ingestion (i.e., 3 mg/kg) and a plyometric based-conditioning activity in enhancing the performance than their separate use. Specifically, 3 mg/kg of CAF is a safe dose and did not result in adverse effects that could impair taekwondo-specific performances. The synergistic effects of caffeine and conditioning activity could be an efficient strategy to prepare athletes to cope with physical and psychological stress during competitions and can serve to organize the pre-competitive routine or training session more effectively.

## Figures and Tables

**Figure 1 nutrients-14-00571-f001:**
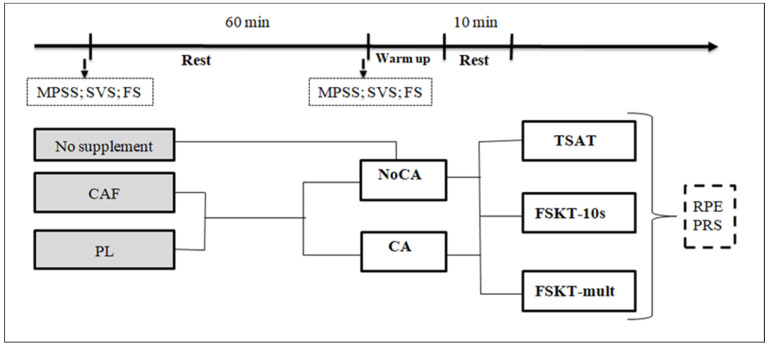
Schematic representation of the study design. MPSS—Mood and Physical Symptoms Scale; SVS—Subjective Vitality Scale; FS—Feeling Scale; RPE—rating of perceived exertion; PRS—perceived recovery status; PL—placebo; CAF—caffeine; CA—conditioning activity; TSAT—taekwondo-specific agility test; FSKT-10s—10 s frequency speed of kick test—FSKT-mult: multiple frequency speed of kick test.

**Table 1 nutrients-14-00571-t001:** Physical performances during taekwondo-specific tests following different conditions (values are mean ± SD; *n* = 20).

		PL + NoCA	CAF + NoCA	PL + CA	CAF + CA	Control	Overall
TSAT (s)	M	5.7 ± 0.5	5.5 ± 0.5	5.3 ± 0.5	5.1 ± 0.4	6.0 ± 0.2	5.5 ± 0.4 ^£^
	F	6.6 ± 0.5	6.3 ±0.4	6.2 ± 0.4	5.8 ± 0.4	6.3 ± 0.5	6.2 ± 0.4
	Overall	6.1 ± 0.5	5.9 ± 0.4 ^α^	5.7 ± 0.5 ^$,§^	5.5 ± 0.4 *^,≠^	6.2 ± 0.3	5.9 ± 0.4
FSKT-10s (*n*)	M	25 ± 2	27 ± 2	27 ± 2	29 ± 2	24 ± 2	27 ± 2d
	F	23 ± 1	24 ± 1 ^b,c^	24 ± 1 ^a^	26 ± 1 *	22 ± 1	24 ± 1
	Overall	24 ± 2	25 ± 1	26 ± 1	28 ± 1 *	23 ± 1	25 ± 1
FSKT-mult (kicks’ number) (*n*)	M	117 ± 7 ^h^	121 ± 8	126 ± 4	138 ± 7 ^f^	107 ± 6 ^g^	122 ± 6 ^d^
	F	105 ± 6	110 ± 10	111 ± 9	119 ± 6 ^f^	95 ± 10	108 ± 8
	Overall	111 ± 6	116 ± 9 *	118 ± 6 *^,e^	128 ± 7 *	101 ± 8	115 ± 7
FSKT-mult (DI) (%)	M	9 ± 2	6 ± 3	7 ± 2	4 ± 1	11 ± 3	7 ± 2 ^d^
	F	10 ± 2	8 ± 2	7 ± 4	6 ± 3	13 ± 4	9 ± 3
	Overall	10 ± 2 ^i^	7 ± 2 ^i^	7 ± 3 ^i^	5 ± 2 *^,§^	12 ± 3	8 ± 2

* Main effect of condition: CAF + CA elicited better performance than CAF + NoCA, PL + NoCA, and control conditions at *p* < 0.001; ^≠^ main effect of condition: CAF + CA elicited better performance than PL + CA at *p* < 0.01; ^$^ main effect of condition: significantly better than PL + NoCA and control at *p* < 0.001; ^§^ main effect of condition: better performance than CAF + NoCA at *p* < 0.05; ^α^ main effect of condition: better than PL + NoCA and control conditions at *p* = 0.004; ^£^ main effect of sex: better performance for males compared with females at *p* = 0.008; ^a^ main effect of condition: better performance in PL + CA compared with PL + NoCA at *p* < 0.01; ^b^ main effect of condition: better performance than PL + NoCA at *p* = 0.005; ^c^ main effect of condition: better performance than control condition at *p* = 0.007; ^d^ main effect of sex: males elicited better performance than females at *p* < 0.001; ^e^: main effect of condition: better performance than PL + NoCA at *p* = <0.05; ^f^: interaction effect between condition and sex: better performance for females in CAF + CA or for males in CAF + CA at *p* < 0.001; ^g^ interaction effect between condition and sex: males in control condition elicited lower values compared with other conditions at *p* < 0.001; ^h^ interaction effect between condition and sex: males elicited higher values than females in PL + NoCA, CAF + NoCA, PL + CA, and CAF + CA at *p* < 0.001; ^i^ main effect of condition: better performance than control condition at *p* < 0.005. TSAT—taekwondo-specific agility test; FSKT-10s—10 s frequency speed of kick test; FSKT-mult—multiple frequency speed of kick test; DI—decrement index; PL—placebo; CAF: caffeine; CA: conditioning activity; *n* = number of techniques.

**Table 2 nutrients-14-00571-t002:** Psychological responses before and after different conditions (values are mean ± SD; *n* = 20).

		Before					After					Overall
		PL + NoCA	CAF + NoCA	PL + CA	CAF + CA	Overall	PL + NoCA	CAF + NoCA	PL + CA	CAF + CA	Overall	
MPSS (a.u.)	M	53 ± 9	48 ± 4	45 ± 6	47 ± 7	48 ± 7	56 ± 9 ^c^	55 ± 6 ^c^	49 ± 6	53 ± 6	53 ± 7	51 ± 7
	F	44 ± 7	49 ± 9	46 ± 2	43 ± 4	45 ± 5	49 ± 9	46 ± 2 ^e^	43 ± 4 ^d^	47 ± 7 ^d^	46 ± 5	46 ± 5
	Overall	48 ± 8	49 ± 6 ^a^	45 ± 4	45 ± 5	47 ± 6	52 ± 9	53 ± 8	46 ± 5	50 ± 6	50 ± 7 ^b^	49 ± 6
SVS (a.u.)	M	29 ± 4	31 ± 3 ^g^	27 ± 5	29 ± 4 ^g^	29 ± 4	28 ± 10	33 ± 4	28 ± 6	34 ± 4	31 ± 6	30 ± 5 ^f^
	F	24 ± 6	27 ± 3	27 ± 2	24 ± 4	26 ± 4	25 ± 9	30 ± 5	27 ± 3	29 ± 3	28 ± 5	27 ± 4
	Overall	26 ± 5	29 ± 3	27 ± 3 ^a^	27 ± 4	27 ± 4	26 ± 10	31 ± 5	28 ± 4	32 ± 4	29 ± 6	28 ± 5
FS (a.u.)	M	2 ± 1	2 ± 1 ^g^	1 ± 2	1 ± 3 ^g^	2 ± 2	2 ± 2	3 ± 2 ^g^	1± 2	4 ± 1 ^g^	2 ± 2	2 ± 2 ^f^
	F	1 ± 1	2 ± 1	1 ± 2	0.3 ± 1	1 ± 1	1 ± 3	2 ± 2	1 ± 3	2 ± 1	2 ± 2	1 ± 2
	Overall	2 ± 1	2 ± 1	1 ± 2	1 ± 2	1 ± 1	1 ± 3	2 ± 2	1 ± 2	3 ± 1	2 ± 2	2 ± 2

^a^ Main effect of condition: different from PL + NoCA and PL + CA at *p* < 0.05; ^b^ main effect of time of measurement; better performance after compared with before at *p* < 0.05; ^c^ interaction effect between sex and time: males elicited higher values before compared with before for PL + NoCA and CAF + NoCA conditions at *p* < 0.05; ^d^ interaction effect between sex and time: females elicited higher values after compared with before for PL + CA and CAF + CA conditions at *p* < 0.05; ^e^ interaction effect between sex and time: females elicited higher values after CAF + NoCA compared with PL + NoCA and PL + CA at *p* < 0.01; ^f^ main effect of sex: males elicited higher values than females at *p* < 0.01; ^g^: interaction effect between sex and condition: higher scores for males than females in CAF + NoCA and CAF + CA conditions at *p* < 0.05. a.u.—arbitrary unit; M—male; F—female; MPSS—Mood and Physical Symptoms Scale; SVS—Subjective Vitality Scale; FS—Feeling Scale; PL—placebo; CAF—caffeine; CA—conditioning activity.

**Table 3 nutrients-14-00571-t003:** Rating of perceived exertion and perceived recovery status scores across different conditions (values are mean ± SD; *n* = 20).

		PL + NoCA	CAF + NoCA	PL + CA	CAF + CA	Control	Overall
RPE (a.u.)	M	7.5 ± 1.5	7.4 ± 0.8	7.1 ± 1.2	7.6 ± 1.0	7.0 ± 1.1	7.3 ± 1.1
	F	6.9 ± 1.5	7.1 ± 1.4	7.9 ± 1.2	7.6 ± 1.2	7.7 ± 1.0	7.4 ± 1.2
	Overall	7.2 ± 1.5	7.25 ± 1.1	7.5 ± 1.2	7.6 ± 1.1	7.4 ± 1.0	7.4 ± 1.2
PRS (a.u.)	M	2.6 ± 0.8	2.7 ± 1.2	2.7 ± 1.5	3.0 ± 1.7	3.2 ± 1.3	2.8 ± 1.3
	F	2.7 ± 1.5	2.7 ± 2.1	2.0 ± 0.7	2.1 ± 0.7	2.8 ± 0.8	2.5 ± 1.2
	Overall	2.7 ± 1.2	2.7 ± 1.6	2.4 ± 1.1	2.6 ± 1.2	3.0 ± 1.1	2.7 ± 1.2

a.u.—arbitrary unit; RPE—rating of perceived exertion; PRS—perceived recovery status; PL—placebo; CAF—caffeine; CA—conditioning activity; M—male; F—female.

## Data Availability

The data presented in this study are available on request from the corresponding authors.

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
