# Peer review of "Acute Effects of Low Dose of Caffeine Ingestion Combined with Conditioning Activity on Psychological and Physical Performances of Male and Female Taekwondo Athletes"

_nutrients, 2022, doi:10.3390/nu14030571_

Round 1

Reviewer 1 Report

This is an excellent paper that adds significantly to our understanding of pre-competition tactics for combat athletes. I was particularly impressed by the readability and translatability for someone who does not have a background in combat sports. I have no significant issues to note.

Author Response

Response to Reviewer 1 Comments

This is an excellent paper that adds significantly to our understanding of pre-competition tactics for combat athletes. I was particularly impressed by the readability and translatability for someone who does not have a background in combat sports. I have no significant issues to note.

Thank you very much for appreciating our work.

We hope that the manuscript has now reached the standard necessary for formal acceptance endorsement in Nutrients.

We look forward to hearing from you.

Best regards

Reviewer 2 Report

This is an interesting paper attempting to describe the role of caffeine on the performance of sport activities in young subjects with a mean age of 17.5 years. Four groups were studied. The effect of low dose of caffeine 3mg/kg alone or combined with conditional activity was compared to placebo group that was subjected or not to conditional activity. Different tests were performed to assess the physical response to caffeine; while scores were used to evaluate the impact on the psychology (mood, vitality, feeling, perceived exertion and recovery) of each of the participants.

In addition, males and females were recruited to help identifying any sex-difference in response to caffeine.

Caffeine combined to conditional activity improve the performance in all subjects. However, both physical and psychological responses were greater in males than in females.

The study is seriously conducted, and the results are convincing. I must congratulate the authors for this well performed study.

Here are few pitfalls that need consideration:

  • Authors can make effort to decrease abbreviations and rend the paper more agreeable or friendly to read
  • Line 168: add the number 4) and 5) to complete the numeration of tests.
  • Line 269: delete “very large”, it is repeated twice

Author Response

Response to Reviewer 2 Comments

The study is seriously conducted, and the results are convincing. I must congratulate the authors for this well performed study.

Thank you very much for appreciating our work.

Point 1: Authors can make effort to decrease abbreviations and rend the paper more agreeable or friendly to read.

Response 1: We thank the expert reviewer for her/his suggestion. To make the paper more readable, a glossary in alphabetical order was added at the end, acronyms were checked for consistency and some were removed.

Point 2: Line 168: add the number 4) and 5) to complete the numeration of tests.

Response 2: Mistake was amended.

Point 3: Line 269: delete “very large”, it is repeated twice.

Response 14: We thank expert reviewer for his suggestion. Mistake was amended.

We hope that the manuscript has now reached the standard necessary for formal acceptance endorsement in Nutrients.

We look forward to hearing from you.

Best regards